# The Future of Precision Oncology

**DOI:** 10.3390/ijms241612613

**Published:** 2023-08-09

**Authors:** Stuart L. Rulten, Richard P. Grose, Susanne A. Gatz, J. Louise Jones, Angus J. M. Cameron

**Affiliations:** 1Prime Omics, Prime Global, Knutsford WA16 8GP, UK; srulten@gmail.com; 2Centre for Tumour Biology, Barts Cancer Institute, Queen Mary University of London, London EC1M 6BQ, UK; r.p.grose@qmul.ac.uk (R.P.G.); l.j.jones@qmul.ac.uk (J.L.J.); 3Cancer Research UK Clinical Trials Unit (CRCTU), Institute of Cancer and Genomic Sciences, University of Birmingham, Edgbaston, Birmingham B15 2TT, UK; s.gatz@bham.ac.uk

**Keywords:** precision oncology, molecular profiling, genomics, transcriptomics, proteomics, microbiome, metabolomics

## Abstract

Our understanding of the molecular mechanisms underlying cancer development and evolution have evolved rapidly over recent years, and the variation from one patient to another is now widely recognized. Consequently, one-size-fits-all approaches to the treatment of cancer have been superseded by precision medicines that target specific disease characteristics, promising maximum clinical efficacy, minimal safety concerns, and reduced economic burden. While precision oncology has been very successful in the treatment of some tumors with specific characteristics, a large number of patients do not yet have access to precision medicines for their disease. The success of next-generation precision oncology depends on the discovery of new actionable disease characteristics, rapid, accurate, and comprehensive diagnosis of complex phenotypes within each patient, novel clinical trial designs with improved response rates, and worldwide access to novel targeted anticancer therapies for all patients. This review outlines some of the current technological trends, and highlights some of the complex multidisciplinary efforts that are underway to ensure that many more patients with cancer will be able to benefit from precision oncology in the near future.

## 1. Introduction

Over the past 20 years, analysis of the molecular landscape that characterizes tumors or disease has driven major advances in understanding the underlying biology of cancer development [1]. As a result, the heterogeneity of cancer is now widely recognized and, for most cancer types, a one-size-fits-all approach to the treatment of patients is becoming obsolete. Instead, precision oncology—treating the right patient with the right treatment at the right time—encompasses a range of rapidly evolving approaches to cancer therapy that take advantage of the ever-increasing knowledge provided by molecular biomarker profiling, with the ultimate aim being that targeted approaches to cancer treatment will result in maximum clinical efficacy, minimal safety concerns, and reduced economic burden [2,3,4].

The ability to profile complex molecular characteristics of patient tumors, such as DNA sequence data, gene expression profiles, protein levels, immune repertoires, and more (Figure 1), has been driven by the evolution of increasingly sophisticated omic methods, which are now showing potential to not only deepen our understanding of cancer biology, but also to become an integral part of clinical practice. This has the potential to revolutionize our understanding of how to treat and manage patients with cancer [5,6,7,8]. Several national genomic initiatives have determined a substantial amount of genetic variation within populations, and the open exchange of molecular data from thousands of cancer samples has increased our understanding of cancer biology, uncovering distinct hallmarks of disease [9,10,11,12,13]. The discovery of new potential targets has driven the development of effective modern drugs and treatment regimens, addressing unmet clinical needs in cancer therapy [14,15,16,17,18]. Taken together, omic technologies offer many opportunities to improve drug discovery, aid clinical diagnostics, and guide decisions for the treatment and management of patients with cancer. However, there are still some challenges to overcome to ensure the success of next-generation cancer therapies. This review highlights recent developments and speculates on potential future applications of precision oncology.

## 2. The Current State of Molecular Profiling in Precision Oncology

### 2.1. Landmark Discoveries in Precision Oncology

Since the coupling of estrogen-receptor expression to the development of certain types of breast cancer in the 1970s [19], our understanding of the biological mechanisms underlying tumor development and patient responses to treatment has increased significantly, enabling the clinical development of a myriad of targeted therapies (Figure 2). Precision oncology approaches include inhibition/blockade of various receptor tyrosine kinases [20,21,22], specific targeting of KRAS proto-oncogene (*KRAS*) mutations [23], synthetic lethal approaches to target homologous recombination deficient (HRD) cancers with poly (ADP-ribose) polymerase (PARP) inhibitors [24], biomarker-driven enhancement of response rates to immune checkpoint blockade [25,26,27], and the generation of chimeric antigen receptor T-cell (CAR-T) therapies [28]. Notable clinical trials this century have demonstrated significant clinical activity with targeted therapeutic approaches for several tumor types, including cetuximab for epidermal growth factor receptor (*EGFR*)-mutant colorectal cancer [22], vemurafenib for B-Raf proto-oncogene (*BRAF*)-mutant melanoma [29], and olaparib for HRD-enriched ovarian cancers [24], as well as gefitinib and sotorasib for non-small cell lung cancers harboring *EGFR* or *KRAS* p.G12C mutations, respectively [20,23]. In addition, biomarker-driven approaches to clinical trials have led to the approval of tumor-agnostic therapies, including pembrolizumab for cancers with microsatellite instability (MSI) [27] and larotrectinib for cancers with neurotrophic tropomyosin-receptor kinase (NTRK) fusions [21].

As molecular technologies have evolved, more complex and cost-effective analysis of biological molecules has become possible, facilitating large genome-, exome-, and phenome-wide association studies and multiparametric (multi-omic) analyses. Recent advances in next-generation DNA sequencing (NGS)-based technologies and bioinformatics have also enabled the analysis of analytes (including genomic DNA as well as circulating tumor DNA (ctDNA), extrachromosomal circulating DNA (eccDNA) and tumor-derived exosomes (TEXs)) in blood, urine, cerebrospinal fluid and other liquid biopsies [42,43,44]. Liquid biopsy assessments offer minimally invasive alternative approaches to the analysis of DNA from tissue biopsies, which have facilitated sampling at multiple points in the patient journey. As exemplified by the Lung TRACERx and other studies, longitudinal monitoring of liquid biopsies can aid early detection of disease relapse, resistance to treatment, and reveal molecular changes occurring during cancer evolution that are actionable for targeted therapies [45,46,47,48].

### 2.2. Comprehensive Diagnostics for Patients with Cancer

As patient responses to targeted therapeutics are not universally similar, a demand has arisen for rapid and cost-effective biomarker diagnostics to determine patients who are most likely to respond to newly developed targeted therapies. Notable examples in clinical use include a range of immunohistochemistry assays for expression of programmed death ligand 1 on tumor and/or immune cells, which are approved for use as companion and complementary tests across a range of tumor types to determine patient eligibility for immunotherapy [49]. A number of molecular diagnostic tests for precision oncology products have also been co-developed to support clinical validation of precision oncology therapeutics and guide treatment decisions (Figure 2). Currently, more than 60 nucleic acid-based tests have been cleared or approved by the US Center for Devices and Radiological Health for oncology indications [37]. Although most approved molecular tests are restricted to single-target assessment within a small number of tumor types, recent advances in NGS technologies have facilitated a shift from single-target diagnostics to comprehensive genomic profiling panels, enabling a one-step testing platform for multiple targeted therapies [50,51]. Initially gaining companion diagnostic approval for lung, skin, and breast cancer indications, NGS-based gene panels are now available for profiling a range of solid tumor types, although significant heterogeneity exists in the availability of testing across different countries [51,52]. A growing number of tumor types are now eligible for whole genome sequencing, from which genomic information can be used to further guide therapeutic decisions (e.g., HRD mutational signatures can guide eligibility for PARP inhibition, while tumor mutational burden (TMB) and MSI status can be used to determine patient eligibility for immunotherapy) [26,27,53]. In addition, NGS-based comprehensive molecular profiling is now common practice for pediatric patients at relapse. Piloted by several molecular profiling studies [54,55,56,57,58], whole genome-, whole exome-, gene panel-, and/or RNA-sequencing are being used within clinical trial settings and, in some countries, have been incorporated into pediatric standard of care [12,59,60,61,62].

### 2.3. Increasing Information to Guide Treatment Decisions

The number of oncology clinical trials has steadily increased over recent years, and the proportion that includes a pharmacogenetic component (i.e., genetic analysis to predict safety outcomes or response to treatment) to patient selection or stratification has remained steady at just over 40%. Robust clinical trial data have led to the approval of a range of clinical molecular tests to guide disease characterization, prognostication, and treatment choice. For example, multigene expression signatures are used to subtype breast cancers and guide clinical decisions on the use of adjuvant chemotherapy or endocrine therapies [36]. New discoveries in disease biology, such as the correlation between genome instability and anti-tumor immune responses [63], the development of novel treatment modalities, such as CAR-T cell therapy [28], and improvements in technologies to couple precision companion diagnostics to targeted therapies [51], have also led to the successful clinical development of a range of precision medicines for patients with cancer. Improved outcomes have been reported in patients across a range of tumor types, including breast, colorectal, lung, ovarian, skin, and hematological cancers [24,28,29,64,65,66,67].

In recent years, tumor-agnostic indications have been approved, based on the assumption that disease development is driven by specific tumor characteristics such as genome instability, inflammation in the tumor microenvironment (TME), or activation of a particular growth signaling pathway, and that these characteristics present common therapeutic targets that transcend the differences between tumors derived from different tissues [21,26,27]. The advent of robust clinical data supporting tumor-agnostic indications presents the challenge of developing companion diagnostic tests that can guide healthcare providers on which of their patients are likely to benefit from precision medicines [68]. Accurate detection of complex genotypic features such as TMB, MSI, or specific gene fusions in a range of tumor types can be challenging [4,50]. However, once reproducible clinical data have been coupled to robust diagnostics, the approval of tumor-agnostic therapies can be expanded to additional populations (e.g., pediatrics) or tumor types (e.g., rare indications) [68].

Despite significant advances in the molecular characterization of a wide range of cancers, and the development of several novel precision oncology therapies, prognostic and predictive assessments for many tumor types are still largely dependent on histopathological or radiological assessments [69,70,71,72], and precision medicine currently remains unavailable for many patients.

## 3. Future Directions for Molecular Profiling of Patients with Cancer

Even though recent trials with targeted therapies have demonstrated encouraging response rates, a consequence of ever-more precise patient selection is that biomarker-positive populations can be very small, and the most effective targeted therapies may only benefit a small number of patients with specific tumor types, while targeted therapies are not yet available for the majority of patients [21,73,74]. Novel approaches to drug development and patient testing, the integration of multi-omic data from preclinical experiments and clinical trials, communication across all disciplines, and global collaboration will ensure that as many patients as possible can benefit from precision-based treatments (Figure 3).

### 3.1. Multi-Omic Profiling in the Characterization of Disease Biology

Profiling of germline DNA and DNA from tumor biopsies has become well established in the field of precision oncology. This approach has been successful in characterizing the mutational landscape of various tumor types, allowing for prognostic assessment as well as identifying novel cancer-related genes that are potential targets for therapy [70,75,76,77]. For example, molecular approaches to tumor profiling have revolutionized the characterization of endometrial cancers, which can now be classified according to DNA polymerase epsilon (*POLE*) mutation status, MSI status, and the number of somatic copy number alterations [77,78]. It is now recognized that the tumor genotype may influence disease prognosis, response to therapy, and risk of recurrence and is being considered in the selection of precise treatments for patients with endometrial cancer [79].

A significant outcome of cancer-focused genomic analyses was the confirmation of significant genetic variation across tumors, not only between different tumor types, but also between tumors with similar histologies from different patients, between primary and metastatic tumors from the same patient, and across different regions of the same primary tumor [1,80,81]. Analysis of blood analytes such as ctDNA may compromise on the evaluation of spatial information, but overcomes some of the challenges of tumor heterogeneity, as circulating tumor-derived molecules may offer a more holistic representation of the tumor landscape in both primary and metastatic disease settings [42,43,44].

Even though molecular profiling has contributed to the development of improved treatment options for patients with certain tumor types, including lung, breast, skin, and colorectal cancers [27,29,65,82], there remains a dearth of biomarkers and targeted therapies for many cancers. For example, pancreatic tumors develop within a dense TME, comprising stromal cells, immune cells, and extracellular matrix. Interactions within the TME can lead to the heterogeneous evolution of many pancreatic cancers, meaning that single targeted therapies may be ineffective [83]. While the existence of distinct pancreatic cancer subtypes is now well described, these are not widely used to guide clinical decision-making [84]. Furthermore, spatial transcriptomics and single-cell studies have indicated that multiple disease subtypes co-exist in most tumors, with clear ramifications for response and resistance to treatment [85,86,87,88]. In addition, actionable mutations in genes including *BRAF* and DNA repair genes have been identified in around half of patients with pancreatic ductal adenocarcinoma using real-time whole-exome sequencing, with second-line targeted therapy showing early promise in selected patients [89]. In pancreatic cancer, it is now widely recognized that combination treatment strategies will likely need to target both the TME and the malignant epithelium to elicit penetrant or durable responses. Consequently, cytotoxic chemotherapy remains the current standard of care for most patients with pancreatic cancers, and efficacy is limited [83]. Transcriptomics offers the opportunity to generate functional tumor profiles, increasing the level of analytical complexity that can be derived from heterogeneous environments [90]. Single-cell RNA sequencing of pancreatic ductal adenocarcinoma biopsies has revealed clinically relevant profiles of both tumor and stromal components [85]. When combined with correlative assessment of bulk RNA sequencing data and meta-analysis of clinical outcomes, the complexities of malignant and stromal compartments in the pancreatic TME can be unraveled, leading to a greater understanding of the disease biology, identification of novel targets, and stratification of targeted treatment regimens [85,91,92].

Applications of proteomic methods to the characterization of cancer continue to evolve. One advantage of proteomics and phosphoproteomics is the ability to provide a direct assessment of the tumor phenotype, rather than using genomic or transcriptomic biomarkers as surrogates to predict drug vulnerabilities [93,94]. Proteomic approaches can be used to detect non-genomic events, including epigenetic changes or influences from the TME, and some preclinical studies have already demonstrated that these approaches may be more reliable predictors of clinical response to treatment than alternative methods [94,95,96]. However, the clinical utility of proteomic markers depends on further retrospective analysis of clinical data to refine hypotheses and the success of their validation through prospective clinical trials [96]. The contribution of proteolytic processes to many aspects of cancer development, through protein degradation, post-translational modification, and cell signaling, is widely reported, and new technologies to assess protease activity and degradomics are already contributing to an increased understanding of cancer biology [97]. Proteomic profiles can be used to train machine learning algorithms to predict patient prognosis and anticancer drug efficacy [98], and increasingly large clinical studies will be performed to develop novel treatments for several solid tumor types and hematological cancers [99,100,101]. Furthermore, optimization of sample preparation and advances in mass spectrometry will enable analysis of the cancer proteome with increased depth and complexity, with larger cohorts adding statistical power to clinical discoveries [99].

### 3.2. Novel Approaches to Drug Development

A key challenge in precision oncology is the expansion of potentially druggable targets [102]. In 2018, collaborators in the Illuminating the Druggable Genome initiative, funded by the US National Institutes of Health, reported that one-third of all human proteins had yet to be studied in detail and that only 3% were targets of at least one approved drug with a known mechanism of action [103]. Integrated analysis of multi-omic, biochemical, and clinical data is likely to support future investigations to determine novel targets and mechanisms of action [70,103]. The challenge remains that, although many kinases, receptors, and receptor ligands have active sites that can be selectively targeted by small molecules or antibodies, many proteins that are potential targets for precision oncology, including transcription factors STAT3, TP53, and MYC, aberrant transcription factor fusion peptides, and several DNA damage response and repair proteins, are not currently druggable by such means [104]. A diversity of oncogenic mutations occurring in a potential target (such as RAS), co-occurring mutations that confer treatment resistance, and loss (rather than gain) of function as the underlying cause of cancer (as seen in HRD cancers) can be challenges to selective inhibition. In cases like these, novel approaches to drug development, such as multifunctional drugs, gene silencing, genetic modification, disruption of the target interactome, target degradation, synthetic lethality, or targeted activation of an antitumor immune response, may be effective [104].

There are now several examples of how bifunctional drugs are showing promise in the development of new anticancer therapies. Bispecific monoclonal antibodies such as amivantamab overcome some of the limitations of monofunctional drugs by simultaneously targeting specific oncogenic factors such as EGFR and drivers of treatment resistance such as mesenchymal-epithelial transition [105]. Other bispecific drugs such as antibody-drug conjugates (ADCs) combine specific target-binding moieties with a disabling method such as cytotoxicity that is alternative to direct target inhibition. For example, the ADC trastuzumab emtansine specifically delivers a microtubule inhibitor to human epidermal growth factor receptor 2 (HER2)-overexpressing cells and has demonstrated enhanced efficacy over trastuzumab alone in patients with HER2-positive breast cancer [106]. In bicycle toxin conjugates (BTCs), antibody species seen in ADCs are replaced with small cyclic peptides, with the aim to improve tissue delivery and pharmacokinetics [107]. Proteolysis-targeting chimeras (PROTACs) promote the degradation of (rather than inhibition of) target proteins by recruiting an E3 ligase and activating polyubiquitination [104]. A number of clinical trials are underway to investigate the safety and efficacy of targeted BTC and PROTAC therapies for cancer [107,108].

Synthetic lethality-based approaches are particularly valuable in the development of novel anticancer therapeutics, and assessment of the genetic variation in a patient’s tumor may uncover specific targets for precision therapies. This approach provides an opportunity to target tumors with mutations in tumor suppressor genes such as *BRCA1*, *BRCA2*, and *RAD51*, where loss of function results in the tumorigenic consequences of deficient homologous recombination [24,109]. In contrast to oncogene-driven cancers, targeted inhibition of the protein products of tumor suppressor genes is not the aim, while restoration of normal protein function is technically complicated. However, targeted inhibition of their synthetic lethal partners can offer considerable efficacy combined with safety profiles that are improved compared with chemotherapeutic alternatives [10,109]. As examples, PARP inhibitors have demonstrated efficacy in the targeted treatment of a range of HRD tumor types [110,111,112], but efforts are underway to identify novel synthetic lethal gene pairs to widen the pool of potential synthetic lethal targets and tumor biomarkers that may predict sensitivity to synthetic lethal agents [10,113].

Drug development may also be facilitated by genomic approaches to determine potential responses to current treatments. CRISPR-Cas9-based mutagenesis studies, including synthetic lethality screens, can rapidly determine which genetic mutations are likely to confer drug sensitivity or resistance [10,114]. Initiatives such as the Cancer Dependency Map integrate data from large numbers of genetic and pharmacological screens, along with the genomic profiles of cellular models to build landscapes of new therapeutic targets, while analysis of large-scale datasets in parallel can be used to estimate the prevalence of mutations that confer sensitivity to those therapies in patient populations [109,115]. In addition, novel therapeutic strategies may include targeting chemically modified neoantigens on the surface of tumor cells, stimulating immune responses against drug-resistant tumors [116].

### 3.3. Clinical Molecular Diagnostics

Cancer biomarkers can be prognostic (associated with disease progression and outcome, irrespective of treatment) or predictive (associated with disease response and survival in patients receiving a particular therapy). Along with the discovery of new biomarkers, the development of novel targeted therapies will increase demand for companion molecular diagnostics to characterize tumors and guide treatment decisions [51]. It is therefore expected that the number of approved molecular diagnostic assays will continue to grow, providing opportunities for precision treatment of patients who currently have limited treatment options [41]. As the cost and turnaround time of molecular approaches such as NGS reduces, comprehensive genomic and transcriptomic profiling of multiple biomarkers in a single sample will become more widely accessible and offer a more rapid alternative to serial biomarker testing by other methods [51,90]. This approach is already having an impact in the characterization of complex lesions such as sarcomas in children as well as adults [117,118,119]. Complex molecular diagnostics with shorter assay times will benefit patients by enabling early detection and characterization of disease, with rapid biomarker evaluation leading to fast precision treatment for patients in urgent need [83]. ctDNA analysis addresses some of the challenges in precision oncology, allowing for minimally invasive testing and convenient, cost-effective methods for screening and early detection of cancer [120]. Increasingly sensitive and sophisticated methods of ctDNA analysis are being developed to address the challenges of variant detection at low allele frequency, allowing earlier detection of oncogenic mutations in blood samples [44,121]. In addition, personalized approaches to ctDNA analysis, which account for the genomic landscape of a patient’s primary tumor, can widen the range of detectable variants in ctDNA, while an increasing number of tumor-agnostic variants are being integrated into non-personalized approaches, where a tumor biopsy is not needed [121]. To date, assessment of ctDNA in pediatric patients has been limited, but the development of novel ctDNA analysis approaches may improve the diagnosis and monitoring of pediatric cancers such as rhabdomyosarcoma [122]. As the range of potential applications of ctDNA analysis increases, increased education and awareness will be needed to ensure that healthcare professionals have rudimentary understanding of the complex bioinformatic flow in ctDNA assessment and of the potential and limitations of each assay, allowing for selection of the most appropriate method of disease assessment [121].

As the demand for molecular testing for precision oncology increases, there is a need to address key challenges in the diagnostic journey. Since accurate diagnosis and disease staging is a priority in cancer diagnostics, tissue sample processing procedures have traditionally been optimized for pathological analysis by histology [123]. This means that there can often be limited tissue available for molecular analyses and that remaining nucleic acids may be damaged by formalin or other fixatives [124]. As the integrity of nucleic acids in patient samples becomes increasingly important, changes in pre-analytic processes will be needed to optimize sample acquisition and processing for nucleic acid analysis. Furthermore, assessment of labile analytes, such as methylated DNA, gene transcripts, or ubiquitinated proteins, will require more precise standardization of many steps in the sample acquisition process, including anesthesia, warm and cold tissue ischemia, macrodissection, molecular extraction, and sample storage [123].

The race for accurate and informative oncology biomarkers has driven a wide variety of analytical outputs in oncology clinical trials, and the need for standardization of molecular diagnostic methods and harmonization of outputs from different assay platforms is widely recognized. Various global initiatives—e.g., those involving the Friends of Cancer Research, Quality in Pathology, and the International Quality Network for Pathology—are underway to ensure quality and reproducibility in the clinical implementation of a number of oncology biomarkers [125,126,127]. Other consortia, including Blood Profiling Atlas in Cancer [128] and CANCER-ID [129], have initiated projects to ensure the standardization of blood-based biomarker analysis.

Access to specialist equipment and analytical expertise can be a barrier to the clinical rollout of any kind of pathological assessment. Patient access to molecular diagnostics can be challenging where expertise in molecular pathology and bioinformatics are limited. Solutions to these limitations are not generalizable and may differ from one region to another [130]. To ensure consistency and accuracy, some molecular profiling assessments may only be available at specialized centers of excellence, whereas the cost of specialized laboratory infrastructures may limit the testing capabilities of local centers [130,131]. However, reductions in the cost of specialist equipment, as well as the development of kit-based genomic, transcriptomic, and proteomic assays with robust bioinformatic algorithms that mitigate the need for specialized expertise, may facilitate the implementation of decentralized molecular diagnostics and increase patient access to testing [130,131,132]. Whether testing is centralized or localized, clinical diagnostics will benefit from the digital exchange of molecular profiling data that can be viewed by a multidisciplinary team of experts (such as molecular tumor boards) to derive accurate and timely diagnoses [5,90]. It is important to note that cancer diagnostics impose a burden on the patient, and continued mindfulness will be required in the age of digital medicine to ensure that patients are tested appropriately, that they receive complementary counselling on the potential implications of the results, and that the security of patient data is guaranteed [5,74,133].

### 3.4. Novel Approaches to Clinical Trials

An increased depth of information that characterizes cancer biology at the molecular level warrants novel approaches to the design of future clinical trials. Basket trials that group patients on the basis of prospective biomarker profiling (rather than tumor type) may improve the statistical power of outcomes in patients with rare tumor types, leading to further approvals of tumor-agnostic indications [74]. New multi-arm multi-stage trial designs will allow for several treatments or hypotheses to be explored in parallel with a shared control arm, leading to efficiency in patient recruitment [134]. Development of novel treatment strategies can be accelerated through the use of molecular assessments that give early indications of response, acting as surrogate endpoints for patient survival [135]. Furthermore, adaptive trial designs that feature Bayesian inferences derived from accumulated data at prespecified interim timepoints will allow for treatment modifications to be introduced midcourse [74,134,136]. Using this model, the addition or extension of promising treatment strategies, and the dropping of unsuccessful ones from clinical trials, can enrich the selection of patients who may benefit from treatment [136].

### 3.5. Guiding Treatment Decisions

A wealth of data is available from trials that may or may not have met their primary endpoints. The next wave of precision oncology depends on retrospective integration of these data with clinical evidence to elucidate biological mechanisms driving response and resistance, and ultimately to predict more accurately which patients are likely to respond to targeted therapies and ensure that each patient is prescribed the most effective therapy for their condition. Key challenges include handling, exchange, and storage of large datasets, as well as interpretation and validation of diverse results from different clinical studies. Global collaboration and standardization of molecular profiling approaches is needed to analyze harmonized clinical outcomes, ensuring robust clinical validation of novel precision medicines [74]. Cloud-based computing and artificial intelligence-based algorithms are developing rapidly to derive predictive models for precision oncology from molecular profiling and clinical data [3,5]. These can be further refined by crowdsourced analysis of publicly available datasets, such as The Cancer Genome Atlas and cBioPortal [137,138,139,140]. The iterative process of retrospective analysis and refining clinical hypotheses can lead to the development of increasingly complex prognostic and predictive biomarker models, featuring new patient eligibility criteria, treatment regimens, and their combinations, which may ultimately improve clinical outcomes [141,142].

Trials in small, defined patient populations can lead to challenges in demonstrating statistically powered real-world evidence that leads to a consensus on the clinical benefit of certain molecular approaches to treatment, ultimately resulting in poor engagement from healthcare and reimbursement providers [5,74]. Improvements in education, awareness, and confidence in the benefits of molecular profiling and data interpretation are likely to increase rates of clinical trial enrolment and referral for treatment, while the development of new therapeutics will provide a wider range of precision treatment options for an increasing number of patients [74,143]. It has, however, become widely recognized that existing clinical trial results and public genomic databases may not be applicable to all patient populations, and efforts are underway to improve equity, diversity, and inclusion in clinical trials, as well as to ensure that reference genomes adequately represent the full range of sequence diversity across human populations [144,145].

In addition to data-guided adaptive designs for clinical trials described above, future cancer treatment strategies may include adaptive schedules that aim to control the growth of drug-resistant tumor cells by maintaining a competitive environment of drug-sensitive cells, thus achieving disease control by reducing the tumor burden to a tolerable level, as opposed to complete eradication [146]. Future treatment guidelines are likely to feature recommendations for biomarker-guided treatment modifications for some tumor types, with treatment intensification ensuring maximal possible benefit for high-risk patients, and treatment deintensification improving quality of life and reducing the economic burden of cancer care for low-risk patients [147]. It is likely that longitudinal disease monitoring will be enhanced by further implementation of biomarker assessment from liquid biopsies. Circulating biomarkers could be used to detect changes during or after therapy to guide adaptive treatment decisions following relapse or recurrence [45,148]. Refinement of liquid biopsy approaches may also provide comprehensive information from heterogeneous tumors or different disease sites, enabling the selection of patients who may benefit from combination therapies that target multiple tumor characteristics through independent mechanisms [44,149].

## 4. Discussion

Recent developments in molecular technologies have enabled the ever more precise and complex characterization of human cancers, while identification of novel targets have, for some patients, delivered of the promise of precision oncology, showing improved response rates while mitigating safety concerns. There is, however, some way to go on the journey to accessible precision medicine for all patients with cancer [150]. Global collaboration is required to establish standardized experimental and reporting methods, as well as clinically relevant cutoffs in biomarker analyses, reducing the noise in data derived from different clinical studies [5,150]. Intelligent approaches to molecular profiling are also required. While comprehensive multi-omic profiling of all patients may one day be a possibility, many are likely to be irrelevant or impractical in a real-world setting, and it is important to ensure that patients undergo the right test for the right biomarkers at the right time. Further efforts are required to ensure that representative molecular datasets are available for all patient populations, that all patients have access to appropriate molecular testing, and that novel targeted approaches are identified for “hard-to-treat” tumor types [74]. Ongoing interdisciplinary communication is needed between product developers, molecular and cell biologists, pathologists, oncologists, governing bodies, and payers, as well as patients and their families, and this can only be achieved with improved awareness and literacy around molecular methods and their applications [143]. It is important to remember that all parties are aware of the strengths and limitations of molecular testing, and that appropriate counseling is available to guide patients and their families through their journey [5]. Finally, we must not forget that ethical guidelines must evolve at the same pace as technological advances, to ensure patient safety and privacy [5,74]. If these challenges are addressed appropriately to inform drug discovery, disease diagnosis, and treatment decisions, improved precision oncology will be made available for a wide range of patients with cancer.

## Figures and Tables

**Figure 1 ijms-24-12613-f001:**
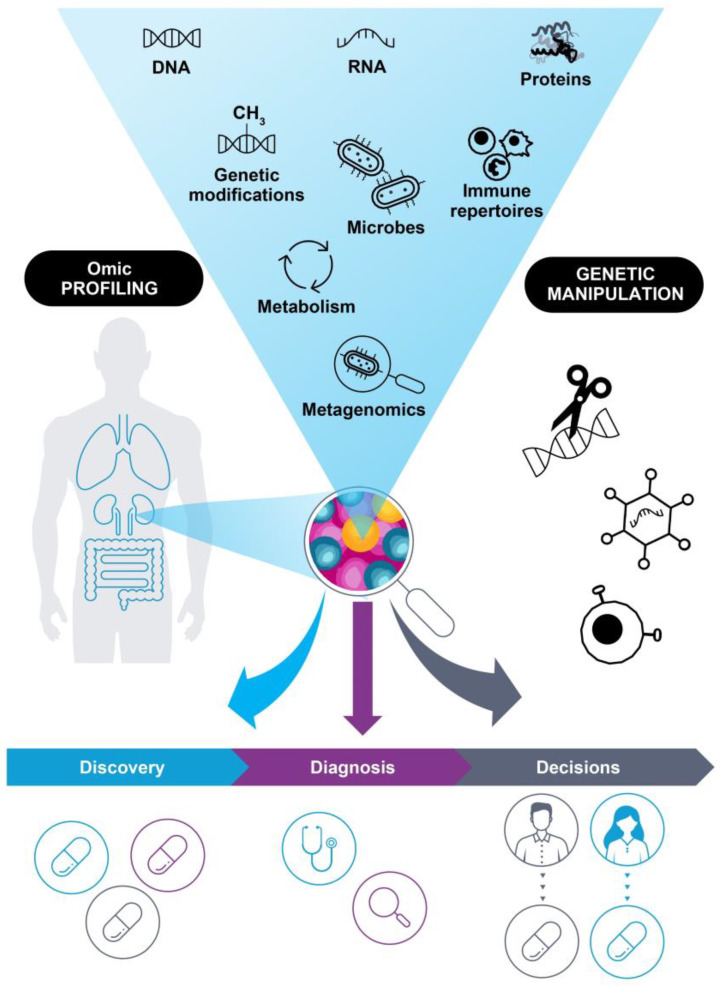
Applications of omic technologies in oncology. Omic profiling has shown increased potential to drive drug discovery, diagnostics, and treatment decisions, due in part to ever-evolving technologies. Characteristic information can be gained from DNA sequencing (genomics), RNA analysis (transcriptomics), protein components (proteomics), analysis of DNA modification (epigenomics), immune repertoires, cellular metabolism (metabolomics), microbial populations living in a patient (microbiomics), and even the study of complex interactions between different organisms and the environment (e.g., metagenomics, metatranscriptomics). In parallel, novel means of genetic manipulation have driven the development of novel therapeutics.

**Figure 2 ijms-24-12613-f002:**
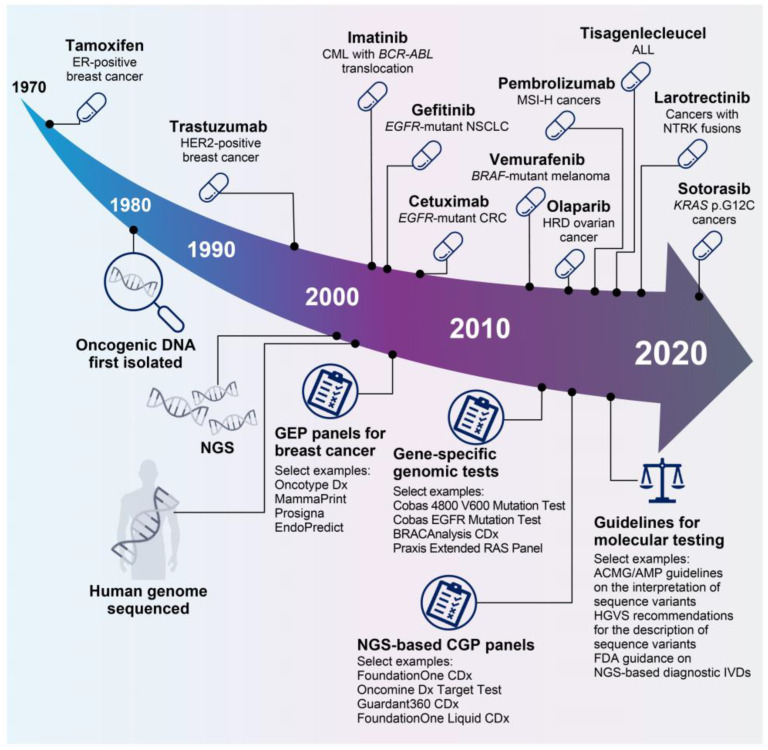
Select key events in 50 years of precision oncology. A deep understanding of the biology underlying cancer development has led to the development of a number of precision oncology therapies [19,20,21,22,24,27,28,29,30,31,32]. In parallel, technological advances such as NGS have facilitated the development of a plethora of precision diagnostics, which have evolved from gene-specific tests to multigene CGP assays [33,34,35,36,37]. In recent years, the need for guidance on molecular testing has been recognized, and a number of guidance notes are now available [38,39,40]. Figure adapted from Colomer et al., 2020 [41] and updated. ABL, tyrosine-protein kinase ABL1; ACMG, American College of Medical Genetics and Genomics; ALL, acute lymphoblastic leukemia; AMP, Association for Molecular Pathology; BCR, breakpoint cluster region protein; BRAF, B-Raf proto-oncogene, serine/threonine kinase; CGP, comprehensive genomic profiling; CML, chronic myeloid leukemia; CRC, colorectal cancer; EGFR, epidermal growth factor receptor; ER, estrogen receptor; FDA, US Food and Drug Administration; GEP, gene expression profiling; HER2, human epidermal growth factor receptor 2; HGVS, Human Genome Variation Society; HRD, homologous recombination deficient; IVD, in vitro diagnostic; KRAS, KRAS proto-oncogene, GTPase; MSI-H, microsatellite instability high; NGS, next-generation sequencing; NSCLC, non-small cell lung cancer; NTRK, neurotrophic tropomyosin-receptor kinase; RAS, GTPase HRas.

**Figure 3 ijms-24-12613-f003:**
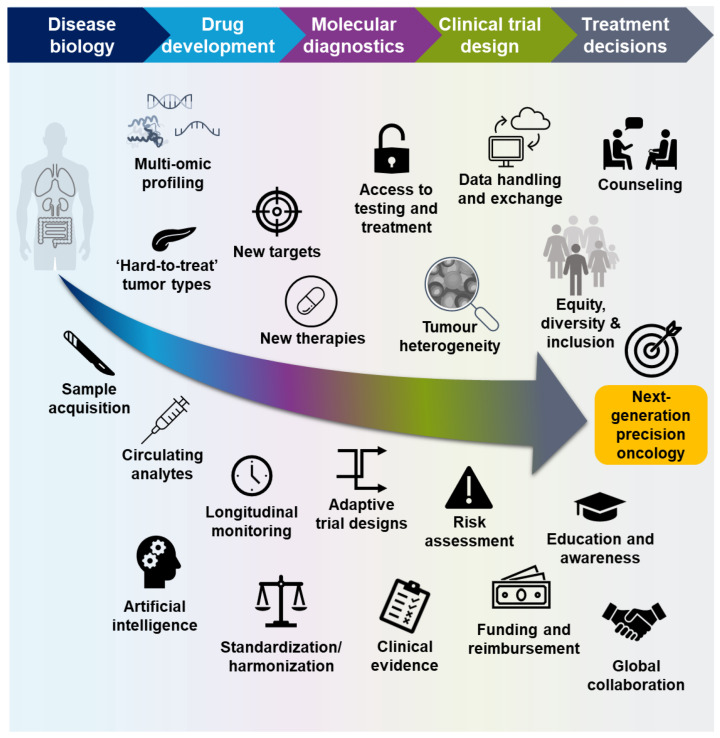
Key steps to ensuring the success of molecular profiling in precision oncology.

## Data Availability

No new data were created or analyzed in this study. Data sharing is not applicable to this article.

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
