# Peer review of "The Future of Precision Oncology"

_ijms, 2023, doi:10.3390/ijms241612613_

Round 1

Reviewer 1 Report

The manuscript by Rulten et al. is a comprehensive review on potential application of omics technologies in oncology. This is an excellent review on this topic, very well-organized and -written, discussing the most important points and raising the questions to be addressed in the future. I enjoyed reading it. Some minor points should be addressed in order to complete this critical revision and provide a more complete review.

1. Page 3 - ‘Figure adapted from Colomer et al., 2020 [41]. …...’ – please transfer to the figure caption.

2. Challenges in liquid biopsy could be described.

3. Clinical trials could be described in paragraph to present the directions taken in precision oncology.

Author Response

We would like to thank the reviewer for the excellent feedback. A point-by-point response to the specific comments can be found below: 1. Page 3 - ‘Figure adapted from Colomer et al., 2020 [41]. …...’ – please transfer to the figure caption.

This has been done. Apologies for the rogue "hard return" in the manuscript formatting that sent part of the figure legend into the main text.

2. Challenges in liquid biopsy could be described.

We have added our thoughts on how the challenges in liquid biopsy assessment can be addressed in the future. We hope the updates in lines 331-334 address this comment sufficiently. We have focused on ctDNA analysis, as we feel this is the furthest along in terms of clinical implementation of blood analyte assessment, but we feel that the suggested solutions could also be applied to other blood analytes.

3. Clinical trials could be described in paragraph to present the directions taken in precision oncology.

We hope that this referred to the key discoveries mentioned in and around Figure 2. We have therefore expanded the tex around Figure 2 to mention several landmark clinical trials this century. Please see lines 76-85 in the revised version for new additions.

Reviewer 2 Report

Since current approaches of cancer treatments are not showing effective clinical outcome in most patients, precision medicine- a treatment approach designed based on the patient’s specific tumor biomarkers, could be the potential future cancer treatment. In the current review, authors provided comprehensive insights about the future of precision medicines in oncology. 

Overall, the review is well written, and authors gave detailed information about use of precision medicine in cancer diagnostics, prognostic, and treatment approaches. However, authors are majorly focused on the targeted therapies. Authors mentioned about cancer immunotherapy in one or two sentences. As Immunotherapy has become standard care for some of the cancers, it would be better to give some insights about use of precision in medicine in immunotherapy of cancer.

Author Response

We thank the reviewer for their feedback.

As immunotherapy has become standard care for some of the cancers, it would be better to give some insights about use of precision in medicine in immunotherapy of cancer.

We agree with the reviewer that precision approaches to immunotherapy warrant further expansion. We have therefore included the use of PD-L1 expression assays (see lines 119-123) as well as TMB and MSI assessment (lines 137-138), which can be used to predict patient responses to immunotherapy.

Reviewer 3 Report

I read with great interest the Manuscript titled " The Future of Precision Oncology”, topic interesting enough to attract readers' attention.

Although the manuscript can be considered already of good quality, I would suggest following recommendations: 

-       I suggest a round of language revision, in order to correct few typos and improve readability.

-       In recent years the classification of endometrial cancer has evolved significantlyit would be interesting to discuss results of this study in the scenario of the current molecular classification and novel insights into molecular mechanisms of endometrial cancer and how could help to determine more accurately prognosis, choose a tailored management and future directions. I would be glad if the authors discuss this important point, referring to PMID: 36833105 and 36979434)

Because of these reasons, the article should be revised and completed. Considering all these points, I think it could be of interest to the readers and, in my opinion, it deserves the priority to be published after minor revisions. 

I suggest a round of language revision, in order to correct few typos and improve readability.

Author Response

We thank the reviewer for this feedback. Please find responses to specific points below:I suggest a round of language revision, in order to correct few typos and improve readability.

The manuscript has now undergone a thorough Editorial QC by professional Editors. Minor updates to the text are highlighted in the revised version.

In recent years the classification of endometrial cancer has evolved significantly, it would be interesting to discuss results of this study in the scenario of the current molecular classification and novel insights into molecular mechanisms of endometrial cancer and how could help to determine more accurately prognosis, choose a tailored management and future directions. I would be glad if the authors discuss this important point, referring to PMID: 36833105 and 36979434)

This is an excellent point and, we agree, worthy of inclusion. We have updated section 3.1 accordingly (see lines 191-198 for additional text).